# Plasma C-Peptide and Risk of Developing Type 2 Diabetes in the General Population

**DOI:** 10.3390/jcm9093001

**Published:** 2020-09-17

**Authors:** Sara Sokooti, Lyanne M. Kieneker, Martin H. de Borst, Anneke Muller Kobold, Jenny E. Kootstra-Ros, Jolein Gloerich, Alain J. van Gool, Hiddo J. Lambers Heerspink, Ron T Gansevoort, Robin P.F. Dullaart, Stephan J. L. Bakker

**Affiliations:** 1Department of Internal Medicine, University Medical Center Groningen, University of Groningen, 9713 GZ Groningen, The Netherlands; l.m.kieneker@umcg.nl (L.M.K.); m.h.de.borst@umcg.nl (M.H.d.B.); r.t.gansevoort@umcg.nl (R.T.G.); dull.fam@12move.nl (R.P.F.D.); s.j.l.bakker@umcg.nl (S.J.L.B.); 2Department of Laboratory Medicine, University Medical Center Groningen, University of Groningen, 9713 GZ Groningen, The Netherlands; a.c.muller@umcg.nl (A.M.K.); j.e.kootstra@umcg.nl (J.E.K.-R.); 3Translational Metabolic Laboratory, Department of Laboratory Medicine, Radboud Institute for Molecular Life Sciences, Radboud University Medical Center, 6525 GA Nijmegen, The Netherlands; Jolein.Gloerich@radboudumc.nl (J.G.); Alain.vanGool@radboudumc.nl (A.J.v.G.); 4Department of Clinical Pharmacy and Pharmacology, University Medical Center Groningen, University of Groningen, 9713 GZ Groningen, The Netherlands; h.j.lambers.heerspink@umcg.nl

**Keywords:** C-peptide, type 2 diabetes, glucose, insulin, hypertension, albuminuria

## Abstract

C-peptide measurement may represent a better index of pancreatic β-cell function compared to insulin. While insulin is mainly cleared by liver, C-peptide is mainly metabolized by kidneys. The aim of our study was to evaluate the association between baseline plasma C-peptide level and the development of type 2 diabetes independent of glucose and insulin levels and to examine potential effect-modification by variables related to kidney function. We included 5176 subjects of the Prevention of Renal and Vascular End-Stage Disease study without type 2 diabetes at baseline. C-peptide was measured in plasma with an electrochemiluminescent immunoassay. Cox proportional hazards regression was used to evaluate the association between C-peptide level and type 2 diabetes development. Median C-peptide was 722 (566–935) pmol/L. During a median follow-up of 7.2 (6.0–7.7) years, 289 individuals developed type 2 diabetes. In multivariable-adjusted Cox regression models, we observed a significant positive association of C-peptide with the risk of type 2 diabetes independent of glucose and insulin levels (hazard ratio (HR): 2.35; 95% confidence interval (CI): 1.49–3.70). Moreover, we found significant effect modification by hypertension and albuminuria (*p* < 0.001 and *p* = 0.001 for interaction, respectively), with a stronger association in normotensive and normo-albuminuric subjects and absence of an association in subjects with hypertension or albuminuria. In this population-based cohort, elevated C-peptide levels are associated with an increased risk of type 2 diabetes independent of glucose, insulin levels, and clinical risk factors. Elevated C-peptide level was not independently associated with an increased risk of type 2 diabetes in individuals with hypertension or albuminuria.

## 1. Introduction

The global number of people with diabetes was estimated 415 million in 2015 and it has been predicted to rise to 642 million by 2040 [1]. Abnormal insulin action, known as insulin resistance (IR) is a main feature underlying development of type 2 diabetes [2]. Hyperinsulinemia as a marker of IR is the result of increased production of insulin by pancreatic β-cells to keep up with increased requirement to maintain near normoglycemia [3]. While secreting increased amounts of insulin, pancreatic ß-cells secrete equimolarly increased amounts of C-peptide [4]. Unlike insulin, of which a considerable portion is cleared by the liver at first pass transit, all synthesized C-peptide reaches the systemic circulation, making it a more accurate marker of endogenous insulin secretion [5,6,7]. Since IR is an important pathophysiological aspect underlying development of cardiovascular disease and even cancer, a considerable number of studies have prospectively investigated a potential association of C-peptide with development of cardiovascular and all-cause mortality [4,5,8,9,10,11]. To the best of our knowledge, only one study to date, prospectively investigated the association of C-peptide with development of type 2 diabetes [6]. Although this latter study only investigated 140 subjects, it suggested a much stronger association for a C-peptide based index than for an index based on insulin concentrations (area under receiver operating characteristic curve of 0.85 versus 0.73). One of the peculiarities of this study was that it excluded subjects with chronic kidney disease, which might have been important because C-peptide is primarily cleared by renal filtration [12]. Therefore, although circulating C-peptide concentrations are generally considered to better reflect insulin secretion than circulating insulin concentrations [13,14,15], this might be more complicated in circumstances of chronic kidney disease. Indeed, one study that investigated potential interaction for the association of C-peptide with outcome, found kidney function to be the only significant effect-modifier among several factors evaluated [4].

The primary aim of our study was to examine the association between C-peptide levels and risk of developing type 2 diabetes, independent of glucose and insulin levels, in a large population-based cohort study with younger age than has been previously evaluated for a long follow-up period. The secondary aim of our study was to examine potential effect-modification by kidney function and factors related to kidney function.

## 2. Materials and Methods

### 2.1. Design and Study Population

The study was based on individuals who participated in the Prevention of Renal and Vascular Endstage Disease (PREVEND) Study, a large cohort study among the general population (age ranged between 28 and 75 years) of the city of Groningen, the Netherlands. The details of this study have been published elsewhere [16]. The baseline cohort was comprised of 8592 individuals who completed the examination in 1997 and 1998. The second screening was performed from 2001 through 2003 with 6894 participants, which is the baseline of the present study.

Of these participants, we excluded 447 individuals with diabetes, 158 subjects with unknown diabetes status, 10 individuals with chronic kidney disease requiring dialysis, 127 subjects who were not studied in the fasting state, 176 individuals with missing values of C-peptide, and 800 subjects with no follow-up data available for diabetes, leaving 5176 participants for the analyses. Evaluation for incident type 2 diabetes took place in three consecutive follow-up screening programs in 2003–2006, 2006–2009, and 2008–2012 respectively.

The PREVEND study was approved by the medical ethics committee of the University Medical Center Groningen and was performed according to the principles outlined in the Declaration of Helsinki guidelines (approval number: MEC96/01/022). Written informed consent was obtained from all participants.

### 2.2. Data Collection

Participants visited the outpatient clinic twice separated by 3 weeks. Prior to the first visit, each participant was asked to fill out a questionnaire regarding demographic data, cardiovascular and renal disease, family history of diabetes, and the information about use of medication that was confirmed by using data from pharmacy registries of all community pharmacies in the city of Groningen [17]. Smoking and alcohol use were based on self-reports. Smoking status was categorized as never, present, and former. Self-reported alcohol consumption was assessed using a question in which participants were asked to choose 1 of the following categories: abstention (no alcohol consumption), 1 to 4 units/month, 2 to 7 units/week, 1 to 3 units/day, or >3 units/day [18]. At the initial visit, body mass index (BMI) was calculated as a ratio of weight (kg) to the height squared (m) after measurement of weight, with a Seca balance scale (Vogel and Halke, Hamburg, Germany), and height to the nearest 0.5 kg and 0.5 cm, respectively. Both in the first and second visit, systolic and diastolic blood pressure was recorded in the supine position, with an automatic Dinamap XL Model 9300 series device. Hypertension was defined by self-reported physician diagnosis, use of antihypertensive medication, or blood pressure ≥140/90 mmHg. Two 24-h urine specimens were collected in the last week before the second visit. At the second screening visit, venous blood samples were drawn from an antecubital vein after a 10-h fast, the assays were performed in EDTA-plasma aliquots that had been stored frozen at −80 °C. 

### 2.3. Laboratory Measurements

High-density lipoprotein (HDL)-cholesterol and triglycerides concentration were measured on a Beckman Coulter AU Analyzer. Plasma aspartate aminotransferase (ASAT), and alanine aminotransferase (ALAT) were measured using the standardized kinetic method with pyridoxal phosphate activation (Roche Modular P, Roche Diagnostics, Mannheim, Germany). Urinary albumin excretion (UAE), given as the mean of the two 24-h urine albumin excretions, was quantified using a nephelometry with a threshold of 2.3 mg/L, and intra- and interassay coefficients of variation of less than 2.2% and less than 2.6%, respectively (Dade Behring Diagnostic, Marburg, Germany). Serum creatinine and cystatin C measurement have been described previously [19]. Estimated glomerular filtration rate (eGFR) was calculated by using the Chronic Kidney Disease Epidemiology Collaboration (CKD-EPI) equation which combines information on serum creatinine and serum cystatin C [20]. Plasma glucose was measured by dry chemistry (Eastman Kodak, Rochester, NY, USA). Fasting insulin was measured with an AxSYM autoanalyzer (Abbott Diagnostics, Amstelveen, The Netherlands). Homeostasis model assessment for insulin resistance (HOMA-IR) was calculated as fasting plasma insulin (µU/mL) × fasting plasma glucose (mmol/L)/22.5 [21]. C-peptide was measured in plasma with an electrochemiluminescent immunoassay, using a Cobas e602 (Roche Modular E, Roche Diagnostics, Mannheim, Germany). The analytical measurement range was 3 to 13,300 pmol/L. The assay has an intra- and inter-assay coefficient of variation of 2.0, 1.7, 1.5% and 3.9, 3.4, 2.8% at levels of 230, 1153, 4877 pmol/L respectively.

### 2.4. Outcome Definition

Incident cases of type 2 diabetes were established if at least one of the following criteria was met: fasting plasma glucose ≥7.0 mmol/L (126 mg/dL); random plasma glucose level ≥11.1 mmol/L (200 mg/dL); self-report of a physician diagnosis; use of glucose-lowering agents based on questionnaire data and central pharmacy registrations [17].

### 2.5. Statistical Analyses

The study population was categorized into sex-specific tertiles of C-peptide concentration. In the baseline study, continuous data were expressed as mean ± standard deviation (SD) or as median and interquartile range (IQR) for normal distribution and skewed distribution of continuous values, respectively. Categorical data were evaluated by means of a chi-squared test. Differences among tertiles were determined by using *p* value for trends or a Kruskal–Wallis test, when applicable. Multivariable linear regression was used to determine the presence of an association between participants’ characteristics with C-peptide levels. In the first model, analyses were adjusted for age and sex, the second model was further adjusted for insulin, and the last model was additionally adjusted for relevant confounding variables.

We applied Cox proportional hazards regression analysis to study the prospective association of C-peptide level with the risk of developing type 2 diabetes. We calculated hazards ratios (HRs) with 95% confidence intervals (CIs) for type 2 diabetes according to base-two logarithmically transformed (Log_2_) C-peptide. Additionally, these associations were evaluated across sex-specific tertiles of C-peptide where the lowest tertile was assigned the reference category. First, we calculated HRs (95% CIs) for the crude model. In model 1, HRs (95% CIs) were calculated after adjustment for age and sex. In model 2, we further adjusted for smoking status and alcohol consumption. In models 3, we further adjusted for BMI, family history of diabetes, and hypertension. In model 4, we calculated HRs (95% CIs) after further adjustment for triglyceride, total cholesterol, and HDL that may be confounders of the association between C-peptide and risk of diabetes. In model 5, we further adjusted for urinary albumin excretion (UAE) and estimated glomerular filtration rate (eGFR). In the last model, we calculated HRs (95% CIs), which were further adjusted for glucose and insulin. The proportional hazards assumption for the models was tested to confirm absence of violation. 

We examined the risk of developing diabetes with and without inclusion of C-peptide for the Framingham Offspring (FOS) risk score by testing for differences in Harrel’s C-statistics and -2 Log likelihood calculated with and without the inclusion of C-peptide in the model [22]. Additionally, to assess the added value of C-peptide, improvement of diabetes prediction was examined in terms of discrimination and integrated discrimination improvement (IDI). This parameter was calculated by subtracting the mean difference of predicted risk between the FOS risk score and the FOS risk score including C-peptide [23].

For secondary analyses, we performed subgroup analyses and evaluated effect modification by sex, age, BMI, eGFR, UAE, glucose, insulin, and hypertension in the analyses of risk of type 2 diabetes by fitting models containing both main effects and their cross-product terms. The differences in association with incident type 2 diabetes by hypertension or UAE were evaluated using subgroup analyses. 

In sensitivity analyses, we applied competing risk analyses based on Fine and Gray’s proportional subhazards model since the death event that happened earlier to type 2 diabetes event could prevent the individuals from type 2 diabetes development. Moreover, we performed logistic regression analyses to evaluate our results without accounting for time to the event.

All *p* values are 2-tailed. A *p* value less than 0.05 was considered statistically significant. All analyses were conducted with the use of the statistical package IBM SPSS (version 23.0.1; SPSS, Chicago, IL, USA) and STATA/SE (version 14; StataCorp., College Station, Texas, USA).

## 3. Results

### 3.1. Baseline Characteristics

The associations between baseline clinical characteristics and plasma C-peptide according to sex-specific tertiles are summarized in Table 1. Baseline median C-peptide was 722 pmol/L (IQR, 566–935 pmol/L), with slightly higher values in men (756 pmol/L; IQR, 284–992 pmol/L) compared with women (691 pmol/L; IQR, 551–887 pmol/L, *p* < 0.001). At baseline, subjects with higher C-peptide levels were more likely to be older, to have a family history of diabetes, to have smoked more frequently in the past, to consume less alcohol, and to have higher BMI and waist circumference. No significant difference among the groups were found in their smoking status, race, and past history of gestational diabetes. Additionally, higher C-peptide was positively associated with higher systolic and diastolic blood pressure, total cholesterol, triglyceride, glucose, insulin, HOMA-IR, ASAT, and ALAT. Men and women in the highest sex-specific tertile of C-peptide used more antihypertensive medications and lipid-lowering drugs. Additionally, these same individuals had lower HDL cholesterol, lower eGFR, higher urinary excretion of albumin, higher creatinine, and higher urea.

### 3.2. Cross-Sectional Associations

The correlations between log-transformed C-peptide levels and variables of interest which are related to type 2 diabetes were analyzed (Appendix A). In the analysis adjusted for age and sex, we found that family history of diabetes, BMI, systolic blood pressure, diastolic blood pressure, total cholesterol, triglyceride, glucose, insulin, plasma ASAT, plasma ALAT, and urinary excretion of albumin were positively and significantly associated with C-peptide levels, whereas use of alcohol, HDL cholesterol, and eGFR were inversely associated with C-peptide levels.

After further adjustment for insulin, the positive association between C-peptide and other variables including BMI, systolic blood pressure, and glucose and inverse association between C-peptide and HDL cholesterol and eGFR remained the same. In the multivariable model which included all the variables, significant positive associations with C-peptide were found for the female sex, BMI, diastolic blood pressure, triglycerides, plasma ALAT, glucose, insulin, and urinary albumin excretion, whereas inverse associations were observed with systolic blood pressure, HDL cholesterol, plasma ASAT, and eGFR. An adjusted R^2^ for the multivariable model was 0.721, with insulin being the most significant contributing determinant of C-peptide (partial R^2^ = 0.169).

### 3.3. Plasma C-Peptide and Type 2 Diabetes

During a median follow-up of 7.2 years (IQR, 6.0–7.7 years), 289 individuals developed type 2 diabetes. The Kaplan–Meier curves for the development of type 2 diabetes according to sex-specific tertiles of C-peptide demonstrated that elevated C-peptide levels are associated with a higher risk of type 2 diabetes, compared to lower C-peptide levels (*p* < 0.001) (Figure 1). Moreover, the association of C-peptide and risk of type 2 diabetes, calculated per doubling (per Log_2_-unit increase) of C-peptide levels and over sex-specific tertiles (Table 2). In the crude model, higher C-peptide level was associated with an increased risk of type 2 diabetes (HR: 6.09; 95% CI: 5.05–7.36). After cumulative adjustment for age, sex (model 1), smoking, alcohol use (model 2), BMI, family history of diabetes, hypertension (model 3), triglyceride, total cholesterol, HDL cholesterol (model 4), eGFR, and urinary albumin excretion (model 5), this association remained statistically significant (HR:3.26; 95% CI: 2.42–4.36). The association remained similar after further adjustment for glucose and insulin in model 6 (HR: 2.35; 95% CI: 1.49–3.70).

The prospective association of C-peptide with incident type 2 diabetes was independent of age, sex, BMI, family history of diabetes, blood pressure, triglycerides, HDL cholesterol, and fasting plasma glucose, all components of the FOS risk score [22]. Moreover, to assess the predictive value of C-peptide for the risk of developing type 2 diabetes, we calculated the Harrell’s C-statistic for the FOS risk score with and without C-peptide. After adding C-peptide to the FOS risk score, the Harrel’s C-statistic of 0.888 (95% CI, 0.870–0.907) for the FOS risk score increased to 0.891 (95% CI, 0873–0.909) with statistically significant improvement (*p* = 0.04, Appendix A). The IDI was significantly changed to 0.006 (*p* = 0.024) after addition of C-peptide to FOS risk score. Investigation of the differences in the -2 Log likelihood of models with and without C-peptide showed that the -2 Log likelihood significantly improved with inclusion of C-peptide (*p* = 0.001).

However, the addition of insulin showed the Harrel’s C-statistic of 0.889 (95% CI, 0.870–0.908), did not significantly improve the Harrell’s C-statistic for the FOS risk score (*p* = 0.27, Appendix A).

### 3.4. Secondary Analyses on C-Peptide and Type 2 Diabetes

To find effect modification, we tested for interaction by gender, age, BMI, eGFR, UAE, glucose, insulin, and hypertension (Appendix A). We found significant effect modification for age, BMI, eGFR, UAE, glucose, insulin, and hypertension if the product term was added to the crude and adjusted for age and sex Cox regression models. In further multivariable Cox regression analyses adjusted for all covariates, the *p* value for interaction remained significant for age (*p* = 0.019), eGFR (*p* = 0.010), UAE (*p* = 0.001), glucose (*p* = 0.015), and hypertension (*p* < 0.001). However, the significant effect modification of age and eGFR disappeared if the product term of those were added with the product term of hypertension at the same time (*p* = 0.35, and *p* = 0.10 for interactions, respectively). Thus, we did further analyses within two subgroups of individuals with and without hypertension. We found that the association between C-peptide and risk of type 2 diabetes was significant in subjects without hypertension after adjustment for age, sex, smoking, alcohol use, BMI, family history of diabetes, hypertension, triglyceride, total cholesterol, HDL cholesterol, eGFR and UAE, glucose and insulin (HR: 3.25; 95% CI: 1.65–6.41), however the association was not significant in individuals with hypertension (HR: 1.81; 95% CI: 0.99–3.25).

The effect modification of albuminuria remained significant, even when the product term of hypertension was added to the last model (*p* = 0.022). Thus, we did further analyses with four subgroups including: UAE < 15 mg/24 h and without hypertension; UAE ≥ 15 mg/24 h and without hypertension; UAE < 15 mg/24 h and with hypertension; UAE ≥ 15 mg/24 h and with hypertension. We found that the association between C-peptide and risk of type 2 diabetes was significant in subgroups of individuals without hypertension both with UAE < 15 mg/24 h and UAE ≥ 15 mg/24 h after adjustment for other covariates in the last model (*p* = 0.011, and *p* = 0.0.35, respectively) (Figure 2).

### 3.5. Sensitivity Analyses on C-Peptide and Type 2 Diabetes

In sensitivity analyses where we restricted the outcome to incident type diabetes, censored for death, we found similar results in our main analyses in all models (Appendix A). In further sensitivity analyses, when we exclude follow-up time from our analyses, the association between C-peptide levels and incident type 2 diabetes remained the same as our main results (Appendix A).

## 4. Discussion

In this large population-based cohort study, C-peptide level was strongly associated with developing type 2 diabetes in men and women without diabetes with a wide age range of 32 to 80 years. Over a median follow-up of 7.2 years, the association remained significant even after adjustment for the risk factors, including BMI, hypertension, glucose, and lipids, as well as adjustment for insulin level, another marker of insulin resistance. While hypertension and albuminuria were effect modifiers, our finding showed among individuals without hypertension and albuminuria, C-peptide level is a reliable marker for the prediction of type 2 diabetes independent of glucose and insulin which are predictors of type 2 diabetes as well [24,25]. 

Overall, our findings are in accordance with those reported previously. In a cross-sectional study of 146 Japanese-American men, they found that higher fasting C-peptide level and greater intra-abdominal fat distribution was associated with the development of non-insulin dependent diabetes mellitus [26]. Furthermore, they found that in multiple regression analyses, the glucose level at 120-min after oral glucose tolerance test (OGTT) and the fasting C-peptide level predicted type 2 diabetes independent of glucose and other clinical factors. In addition, in a prospective study of 140 individuals without diabetes, C-peptide based index ((C-peptide 30 min − C-peptide 0 min)/(glucoses 30 min − Glucose 0 min]) was an independent predictor for incident type 2 diabetes after adjustment for HbA1c, OGTT, and insulinogenic index [6]. We found similar results in multivariable Cox regression analyses, adjusted for more variables including lipid profile, hypertension, family history of diabetes, and kidney parameters in a larger population with both genders. In a cross-sectional study including 420 individuals, both fasting serum insulin and C-peptide levels were associated with diabetes that increased progressively from individuals with normal glucose tolerance to impaired fasting glucose, impaired glucose tolerance and type 2 diabetes, respectively. Fasting C-peptide was a strong predictor of newly subjects diagnosed with type 2 diabetes independent of BMI, waist circumference, and insulin (OR: 10.85; 95% CI: 2.27–51.8) [27]. Similarly, we found that C-peptide is associated with development of type 2 diabetes independent of insulin and other metabolic factors in a large population study. Our study demonstrated that increased C-peptide level could be helpful to predict type 2 diabetes in the early stages before the signs and symptoms of type 2 diabetes are met. This regards to the pathophysiological changes that take place earlier in the body in term of insulin resistance status. While insulin secretion is dependent on β-cell response to plasma glucose level, the β-cell response to plasma glucose level changes gradually [21,28]. Higher insulin level, as well as higher C-peptide level, would be detected to overcome the higher glucose level. In healthy subjects, pancreatic β-cells secrete relatively constant levels of insulin and C-peptide. Therefore, the fasting condition could be used to represent the basal levels of those markers. However, in individuals with diabetes and impaired glucose tolerance, the correlation between fasting insulin and insulin resistance is as strong as individuals with normal glucose tolerance [29,30]. Of further interest, we found a relatively high frequency of individuals that reported to consume no alcohol among individuals with high fasting C-peptide concentrations, which was accompanied by a relatively low frequency of individuals that reported to consume between 2 units of alcohol per week and 3 units of alcohol per day. This is consistent with earlier reports on an inverse association between intake of alcohol and C-peptide concentration reported in both men and women [31,32]. 

In our study, we confirmed the fasting C-peptide level ability to predict type 2 diabetes by significant improvement of the FOS risk score for prediction of diabetes after it was added to the FOS risk score [22]. On the other hand, insulin was not able to improve type 2 diabetes prediction when it was added to the FOS risk score. Interestingly, this is the first study showing that C-peptide addition to the FOS risk score is able to identify individuals at risk of type 2 diabetes. Even though the effect size of fasting C-peptide was relatively small, these results suggest that C-peptide as a marker of insulin resistance status could be accounted as a new risk factor for prediction of type 2 diabetes.

Our finding showed that elevated C-peptide is strongly associated with increased risk of type 2 diabetes in individuals without hypertension, whereas in the hypertension subgroup, we could not find that association. Participants with hypertension have higher levels of C-peptide in comparison with those without hypertension. That would be consistent with the previous finding reporting the association between insulin resistance or hyperinsulinemia with hypertension [33,34,35]. The underlying pathogenesis is the role of insulin for salt reabsorption and impaired vasodilation which is not related to glucose metabolism [36,37,38]. This confirms that higher C-peptide in individuals with hypertension is independent of impaired glucose metabolism and cannot predict type 2 diabetes. Furthermore, among people with hypertension, C-peptide ability to predict type 2 diabetes in individuals with UAE ≥ 15 mg/24 h becomes worse compared with subjects UAE <15 mg/24 h. This would be justified by the association between insulin resistance and microalbuminuria in patients with essential hypertension and prediabetes [39,40]. The interaction between C-peptide and albuminuria was more remarkable than the interaction between C-peptide and eGFR. Although eGFR is one of the most important markers of kidney function, albuminuria demonstrated to have strong association with kidney dysfunction disease as well as its progression [41]. Moreover, the complex association of albuminuria and hypertension has been focused on their relationship in two ways: on one hand, albuminuria is a sign of renal damage and on the other hand, subclinical renal damage may precede and predict the onset of hypertension and outcomes [42]. C-peptide is mostly metabolized by the kidneys and renal extraction of C-peptide accounts for approximately 85% of the total metabolic clearance [13]. Thus, C-peptide is a marker which could be affected by renal damage which most probably occurs in people with albuminuria and hypertension.

The relationship between insulin resistance measured by C-peptide in individuals without diabetes and future death was previously evaluated in two studies driven from the Third National Health and Nutrition Examination Survey (NHANES-3) [4,5]. The increased risk of cardiovascular or overall death was detected in participants with the highest C-peptide quartile independent of diabetes or other cardiovascular disease risk factors. Moreover, serum C-peptide was significantly associated with risk of death independent of clinical characteristics and HOMA-IR in individuals without diabetes (HR 1.36, 95% CI 1.08–1.70). These studies serve an important message that C-peptide is a marker of insulin resistance state in individuals without diabetes to predict future outcome in terms of mortality. Similarly, when we performed competing risk analyses with mortality in our population, the association between C-peptide level and incident type 2 diabetes remained same as our main results, suggesting that participants without diabetes with a higher C-peptide level were found to have more features of insulin resistance and were at higher risk of developing type 2 diabetes independent of risk factors for type 2 diabetes and cardiovascular disease even when death event was excluded.

Nowadays, according to the World Health Organization (WHO) guidelines, the role of C-peptide in the diagnose of diabetes is only limited to provide help to distinguish unknown cases of type 1 diabetes or type 2 diabetes [43]. The fact that we found fasting plasma C-peptide levels to independently predict development of type 2 diabetes could be used as an argument to incorporate fasting plasma C-peptide in risk prediction models. However, addition of fasting plasma C-peptide levels to the FOS risk score only modestly improves the C-statistic of the score from 0.888 to 0.891 is congruent with its limited value from a clinical point of view.

The strength of our study is a large number of participants with a varied age in a long time duration follow-up. In addition, eGFR and albuminuria were measured which can affect the C-peptide metabolization. Our study’s limitation was that the majority of the individuals in our study were Caucasian, precluding extrapolation to other ethnicities. Nonetheless, our results are in line with previous studies which were performed in the population with different ethnicities [6,25,26].

## 5. Conclusions

In conclusion, in this population based-cohort study, elevated C-peptide level is associated with an increased risk of type 2 diabetes independent of glucose, insulin levels, and other clinical factors in the general population. Elevated C-peptide level is not independently associated with an increased risk of type 2 diabetes in individuals with hypertension or albuminuria.

## Figures and Tables

**Figure 1 jcm-09-03001-f001:**
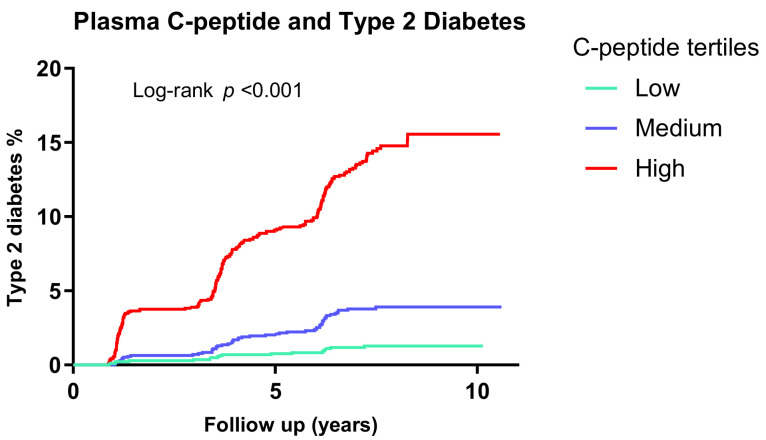
Kaplan–Meier curves for diabetes survival according to sex-specific tertiles of C-peptide in 5176 participants of the PREVEND study. Low = M: <642, F: <592 pmol/L; medium = M: 642–890, F: 592–803 pmol/L; high = M: >890, F: >803 pmol/L; PREVEND: Prevention of Renal and Vascular End-Stage Disease.

**Figure 2 jcm-09-03001-f002:**
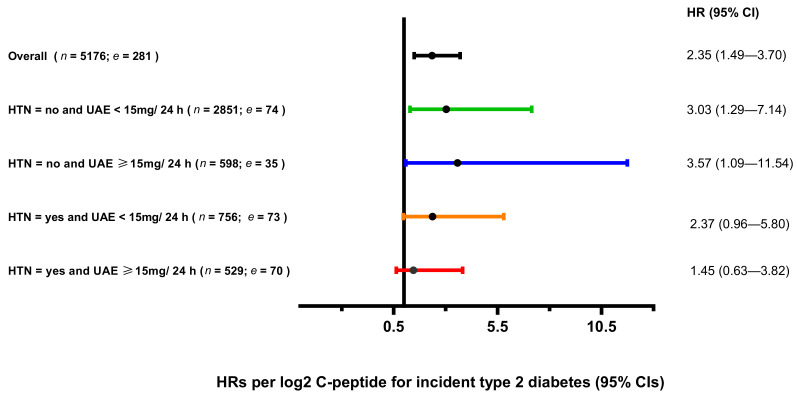
Association between C-peptide and risk of type 2 diabetes in four subgroups. Multivariable hazard ratios (95% confidence intervals) for risk of diabetes are expressed per Log_2_-unit increase of C-peptide levels (*n* = number of subjects; *e* = number of events). Hazard ratios (95 CIs) were derived from Cox proportional hazards regression models with adjusted for age, sex, smoking status, alcohol consumption, BMI, family history of diabetes, triglycerides, total cholesterol, HDL cholesterol, eGFR, urinary albumin excretion, glucose, and insulin. HTN: hypertension; UAE: urinary albumin excretion; BMI: body mass index; HDL: high-density lipoprotein; eGFR: estimated glomerular filtration rate.

**Table 1 jcm-09-03001-t001:** Baseline characteristics according to sex-specific tertiles of C-peptide in 5176 participants of the Prevention of Renal and Vascular End-Stage Disease (PREVEND) study.

	Sex-Specific Tertiles of C-Peptide pmol/L				
	Male	<642	642–890	>890	*p* value for trend *
	Female	<592	592–803	>803	
Participants, *n*		1723	1725	1728	
Female, (%)		50.3%	50%	50.3%	0.986
Age		48.7 ± 10.6	52.3 ± 11.4	56.5 ± 11	<0.001
Race, white (%)		99	99.1	99.5	0.123
The family history of diabetes (%)		13.8	17.6	21.8	<0.001
Smoking status,					
Never (%)		34.3	29.6	26.1	<0.001
Current (%)		27.0	28.4	25.3	
Former (%)		38.7	42.0	48.6	
Alcohol consumption,					
none (%)		18.9	22.6	28.9	<0.001
1–4 units per month (%)		15.8	18.2	17.8	
2–7 units per week (%)		35.3	32.7	29.2	
1–3 units per day (%)		25.2	22.9	19.8	
>3 units per day (%)		4.8	3.5	4.4	
Gestational Diabetes (%)		1.0	2.2	1.6	0.337
Length (cm)		174 ± 9	173 ± 9	171 ± 9	<0.001
Weight (kg)		72.9 ± 11.3	78.5 ± 12.3	86.7 ± 15.1	<0.001
BMI (kg/m^2^)		23.9 ± 2.7	26.1 ± 3.2	29.3 ± 4.3	<0.001
Systolic blood pressure (mmHg)		118.9 ± 15.8	124.1 ± 17	131.7 ± 18.2	<0.001
Diastolic blood pressure (mmHg)		70.3 ± 8.7	73.1 ± 8.6	75.8 ± 8.6	<0.001
Use of antihypertensive medication (%)		6.5	12.2	27.3	<0.001
Hypertension (%)		14.7	23.4	44.3	<0.001
Total Cholesterol (mmol/L)		5.1 ± 0.9	5.4±1	5.6±1	<0.001
HDL-cholestrerol (mmol/L)		1.37 ± 0.30	1.26 ± 0.26	1.15 ± 0.27	<0.001
Triglycerides (mmol/L)		0.85 (0.64–1.11)	1.08 (0.83–1.49)	1.47 (1.08–2.03)	<0.001
Use of lipid-lowering medication (%)		3	6.7	14.2	<0.001
Glucose (mmol/L)		4.5 ± 0.5	4.8 ± 0.5	5 ± 0.6	<0.001
Insulin (mU/L)		5.2 (4.2–6.6)	7.8 (6.5–9.5)	13.1 (10.3–17.9)	<0.001
HOMA-IR ((mU mmol/l2)/22.5)		1.1 ± 0.5	1.8 ± 0.8	3.4 ± 1.9	<0.001
Plasma ASAT (U/L)		22 (19–25)	22 (19–26)	24 (20–28)	<0.001
Plasma ALAT (U/L)		15 (12–20)	17 (12–23)	21 (15–29)	<0.001
Plasma urea (mmol/L)		4.9 ± 1.2	5.1 ± 1.3	5.5 ± 1.6	<0.001
eGFR (mL/min/1.73 m^2^)		99.1 ± 13.7	94.1 ± 15.4	86.9 ± 17.3	<0.001
UAE (mg/24 h)		7.52 (5.73–11.15)	8.26 (6.00–13.27)	10.19 (6.56–20.91)	<0.001

Continuous variables are reported as mean ± SD or median (interquartile range) and categorical variables are reported as percentage. * Determined by linear-by-linear association chi-square test (categorical variables) and linear regression (continuous variables). BMI: body mass index; HDL: high-density lipoprotein; HOMA-IR: Homeostatic model assessment of insulin resistance; ASAT: L-aspartate aminotransferase; ALAT: L-alanine aminotransferase; eGFR: estimated glomerular filtration rate; UAE: urinary albumin excretion.

**Table 2 jcm-09-03001-t002:** Association between C-peptide and risk of developing Type 2 Diabetes in 5176 participants of the PREVEND study *.

	Sex Specific Tertiles of Plasma C-Peptide, pmol/L				C-Peptide Per Log_2_ Unit Increase	
	Male	<642	642–890	>890		*p* value
	Female	<592 1	592–803 2	>803 3		
Cases		19	57	213	289	
Person-years		11,860	11,643	10,756	34,260	<0.001
Crude analysis		1.00 (ref)	3.05 (1.81–5.13)	12.45 (7.78–19.91)	6.09 (5.05–7.36)	<0.001
Model 1		1.00 (ref)	2.83 (1.68–4.77)	10.77 (6.69–17.33)	5.47 (4.48–6.68)	<0.001
Model 2		1.00 (ref)	2.78 (1.65–4.68)	10.41 (4.46–16.75)	5.38 (4.35–6.51)	<0.001
Model 3		1.00 (ref)	2.16 (1.28–3.66)	5.21 (3.15–8.61)	3.47 (2.72–4.43)	<0.001
Model 4		1.00 (ref)	2.01 (1.16–3.50)	3.97 (2.30–6.85)	2.90 (2.20–3.80)	<0.001
Model 5		1.00 (ref)	1.93 (1.1–3.37)	3.75 (2.16–6.52)	3.26 (2.42–4.36)	<0.001
Model 6		1.00 (ref)	1.65 (0.94–2.89)	2.40 (1.32–4.36)	2.35 (1.49–3.70)	<0.001

* HRs (95% CIs) were derived from design-based Cox proportional hazard models. Model 1 is adjusted for age and sex; model 2 is additionally adjusted for smoking status and alcohol consumption; model 3 is additionally adjusted for BMI, family history of diabetes, and hypertension; model 4 is additionally adjusted for triglycerides, total cholesterol, and HDL cholesterol; model 5 is additionally adjusted for eGFR and urinary albumin excretion; model 6 is additionally adjusted for glucose and insulin. eGFR: estimated glomerular filtration rate; PREVEND: Prevention of Renal and Vascular End-Stage Disease.

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
