# Peer review of "Plasma C-Peptide and Risk of Developing Type 2 Diabetes in the General Population"

_jcm, 2020, doi:10.3390/jcm9093001_

Round 1
Reviewer 1 Report
This is necessary and timely research. Overall the paper is soundly constructed and reported.
With respect to alcohol consumption, the results are opposite from that expected. Could the methodology please clarify whether the "alcohol consumption, none" means "never consumed", or a "history of consumption, but none for the last xxx years/weeks" Further discussion on the inverse relationship between the alcohol consumption and higher c-peptide levels would be appreciated.
Line 105, it is unclear how eGFR was calculated. Was it a specific equation from the CKD-EPI collaboration?
However, the presentation is marred by numerous spelling and grammatical errors. For example, "diastolic" is spelt both "Dystolic" and "diastolic". C-peptide and log-liklihood are more often presented as sentence case as neither is a proper noun. With the frequency of use of words with hyphens, using a "nonbreaking hyphen" would aid readability.
Thank you for the opportunity to review this paper.
Author Response
Response to Reviewer 1 Comments
This is necessary and timely research. Overall the paper is soundly constructed and reported.
Response. We would like to thank the reviewer for the thoughtful insights and remarks.
Point 1. With respect to alcohol consumption, the results are opposite from that expected. Could the methodology please clarify whether the "alcohol consumption, none" means "never consumed", or a "history of consumption, but none for the last xxx years/weeks" Further discussion on the inverse relationship between the alcohol consumption and higher c-peptide levels would be appreciated.
Response 1. We agree with the reviewer that the results of the association between alcohol consumption and C-peptide levels were opposite from that expected. Alcohol consumption, none means alcohol is not currently used, which is based on self-report by the participants. To accommodate the comment of the reviewer, we added mentioning of the categories of alcohol intake to the revised version of the manuscript (in lines 91-94). We also looked at the association between use of alcohol and C-peptide levels in individuals using alcohol within the 5 groups of alcohol consumption according to the questionnaire data which are presented in Table 1. In addition, we have included a paragraph on the association of C-peptide with alcohol intake the discussion section of the revised version of the manuscript in lines 311-316.
Point 2. Line 105, it is unclear how eGFR was calculated. Was it a specific equation from the CKD-EPI collaboration?
Response 2. We indeed used a specific equation from the CKD-EPI collaboration. To make this more clear we have rewritten the sentence in which the use of this equation is described (see line 115 of the revised version of the manuscript).
Point 3. However, the presentation is marred by numerous spelling and grammatical errors. For example, "diastolic" is spelt both "Dystolic" and "diastolic". C-peptide and log-liklihood are more often presented as sentence case as neither is a proper noun. With the frequency of use of words with hyphens, using a "nonbreaking hyphen" would aid readability.
Response 3. We have corrected spelling and grammatical errors in the revised manuscript.
.

Reviewer 2 Report
The authors evaluate the potential association between fasting plasma C-peptide and type 2 diabetes (T2D) development in 5176 persons without diabetes of the PREVEND cohort, after a median follow-up of 7.2 yrs (IQR: 6.0-7.7). They perform an elegant and comprehensive statistical analysis, showing that baseline higher plasma C-peptide levels are independently associated with higher risk of T2D developing, after adjustments for age, sex, alcohol consumption, smoking, BMI, family history of T2D, hypertension, triglycerides, HDL cholesterol, eGFR, albuminuria and fasting glucose and insulin (main objective). They also try to show that fasting plasma C-peptide levels may improve the prediction of T2D obtained with the Framingham Offspring (FOS) risk score which includes age, sex, BMI, family history of T2D, blood pressure, triglycerides, HDL cholesterol and fasting plasma glucose for its calculation. They also report that the association between higher plasma C-peptide levels and T2D development is stronger in normotensive and normoalbuminuric individuals and is absent is those with hypertension or albuminuria (subgroup analyses). The authors assess insulin resistance (IR) with the HOMA-IR index which include fasting plasma glucose and fasting insulin for its calculation. They suggest that higher baseline fasting plasma C-peptide may reflect higher insulin IR levels, which would be the pathophysiological basis for the association between these levels and further development of T2D.
Strengths: 1) A cohort with a large number of subjects with a long follow-up period. 2) Assessment of eGFR and albuminuria, factors that may modify plasma C-peptide levels. 3) Comprehensive statistical analyses.
Limitations to be taken into account and properly addressed:
- The addition of fasting plasma C-peptide levels to the FOS risk score improves the C-statistic of the score from 0.888 to 0.891 (p = 0.04) that would be of limited value from a clinical point of view. I think that this fact should be reported in the paper. For example, lines 339-346 should be rewritten. Demonstrating an independent association with T2D development usually does not translate in a clinical significant improvement in the prediction of T2D development, as it was apparently the case
- Because results obtained for hypertensive and normotensive subjects were obtained from subgroup analyses, the last line of the manuscript (lines 354-355) seems to be very assertive and should be softened. This would apply for lines 35-36 in the abstract too.
Additional comments:
- Some information on how often participants were evaluated for T2D development should be provided.
- If authors are interested in evaluating the potential role of IR in T2D development, I would suggest the use of plasma C-peptide levels in the HOMA2 model (available at: https://www.dtu.ox.ac.uk/homacalculator/). This strategy would even allow to explore a novel alternative model to the FOS risk score.
- I would suggest comparing C-peptide against insulin in terms of C-statistic when added any of them to the FOS risk score.
- I suggest for improving the understanding of figure 2 to provide HR (CI 95%) values
- Figure S1: I think is more informative to provide HR(CI 95 %) values than p values.
Author Response
Response to Reviewer 2 Comments
The authors evaluate the potential association between fasting plasma C-peptide and type 2 diabetes (T2D) development in 5176 persons without diabetes of the PREVEND cohort, after a median follow-up of 7.2 yrs (IQR: 6.0-7.7). They perform an elegant and comprehensive statistical analysis, showing that baseline higher plasma C-peptide levels are independently associated with higher risk of T2D developing, after adjustments for age, sex, alcohol consumption, smoking, BMI, family history of T2D, hypertension, triglycerides, HDL cholesterol, eGFR, albuminuria and fasting glucose and insulin (main objective). They also try to show that fasting plasma C-peptide levels may improve the prediction of T2D obtained with the Framingham Offspring (FOS) risk score which includes age, sex, BMI, family history of T2D, blood pressure, triglycerides, HDL cholesterol and fasting plasma glucose for its calculation. They also report that the association between higher plasma C-peptide levels and T2D development is stronger in normotensive and normoalbuminuric individuals and is absent is those with hypertension or albuminuria (subgroup analyses). The authors assess insulin resistance (IR) with the HOMA-IR index which include fasting plasma glucose and fasting insulin for its calculation. They suggest that higher baseline fasting plasma C-peptide may reflect higher insulin IR levels, which would be the pathophysiological basis for the association between these levels and further development of T2D.
Strengths: 1) A cohort with a large number of subjects with a long follow-up period. 2) Assessment of eGFR and albuminuria, factors that may modify plasma C-peptide levels. 3) Comprehensive statistical analyses.
Response. We would like to thank the reviewer for the thoughtful insights and remarks.
Limitations to be taken into account and properly addressed:
Point 1. The addition of fasting plasma C-peptide levels to the FOS risk score improves the C-statistic of the score from 0.888 to 0.891 (p = 0.04) that would be of limited value from a clinical point of view. I think that this fact should be reported in the paper. For example, lines 339-346 should be rewritten. Demonstrating an independent association with T2D development usually does not translate in a clinical significant improvement in the prediction of T2D development, as it was apparently the case
Response 1. We agree with the reviewer about the limited value of C-peptide measurement in diabetes prediction from a clinical point of view. To accommodate the comment of the reviewer, we have rewritten the concerned sentences in lines 365-369.
Point 2. Because results obtained for hypertensive and normotensive subjects were obtained from subgroup analyses, the last line of the manuscript (lines 354-355) seems to be very assertive and should be softened. This would apply for lines 35-36 in the abstract too.
Response 2. We agree with the reviewer and softened the tone of the concerned sentences in lines 379-380 and line 36 of the abstract.
Additional comments:
- Some information on how often participants were evaluated for T2D development should be provided.
Response 1. Accordingly, information about how often participants were evaluated for the development of T2D has been added to the methods section in lines 80-82.
- If authors are interested in evaluating the potential role of IR in T2D development, I would suggest the use of plasma C-peptide levels in the HOMA2 model (available at: https://www.dtu.ox.ac.uk/homacalculator/). This strategy would even allow to explore a novel alternative model to the FOS risk score.
Response 2. We thank the reviewer for the suggestion. The aims of our study were to examine the association between C-peptide levels and risk of developing T2D, independent of glucose and insulin levels and to examine potential effect modification by kidney function and factors related to kidney function. The role of IR in T2D development is established. Because it was not one of our aims to evaluate the role of IR in T2D development and the role of IR in T2D development is already established we refrained from these analyses.
- I would suggest comparing C-peptide against insulin in terms of C-statistic when added any of them to the FOS risk score.
Response 3. We thank reviewer for the suggestion, because it makes us aware of the fact that we made not clear enough that this analysis was performed. To accommodate the comment of the reviewer, and to more clearly present the results of these analyses, we made an additional supplementary table in which the results of the Harrell’s C-statistic analyses are presented and added mentioning of this additional supplementary table to the results section of the revised version of the manuscript (lines 235-237).
- I suggest for improving the understanding of figure 2 to provide HR (CI 95%) values
Response 4. We agree with the reviewer that HR (CI 95%) is informative. Accordingly, in the revised manuscript we have now added HR (CI 95%) to figure 2.
- Figure S1: I think is more informative to provide HR(CI 95 %) values than p values.
Response 5. We agree with the reviewer that presenting of HR (CI 95%) is most informative. However, please note that the p-values in this figure are P-values for interaction. Accordingly, we have left these p-values in the figure and have now added HR (CI 95%) for each subgroup in figure S1.

Reviewer 3 Report
This study assessed serum C peptide levels as a prognostic marker for the evaluation of type 2 diabetes development. They found that it was an effective predictor of type 2 diabetes development in individuals without hypertenison/albuminuria.
This is a very well conducted and written study. My one minor comment would be on line 92-93 to specify the vein used for venous blood samples and additional information regarding storage (i.e spun and stored as plasma? anti-coagulant used? etc) in case these factors affect performance of assays used.
Author Response
Response to Reviewer 3 Comments
This study assessed serum C peptide levels as a prognostic marker for the evaluation of type 2 diabetes development. They found that it was an effective predictor of type 2 diabetes development in individuals without hypertenison/albuminuria.
Point 1. This is a very well conducted and written study. My one minor comment would be on line 92-93 to specify the vein used for venous blood samples and additional information regarding storage (i.e spun and stored as plasma? anti-coagulant used? etc) in case these factors affect performance of assays used.
Response 1. We appreciate the kind words of the reviewer. Venous blood was obtained from an antecubital vein. The baseline assays were performed in EDTA-plasma aliquots that had been stored frozen at -80°C. We have added this information to lines 101 -103.
